# Ethnic Minorities’ Experiences of Cardiac Rehabilitation: A Scoping Review

**DOI:** 10.3390/healthcare11050757

**Published:** 2023-03-04

**Authors:** Aiesha Carew Tofani, Elaina Taylor, Ingrid Pritchard, Jessica Jackson, Alison Xu, Yasuhiro Kotera

**Affiliations:** 1College of Health, Psychology and Social Care, University of Derby, Derby DE22 1GD, UK; 2School of Health and Social Care, Swansea University, Swansea SA2 8PP, UK; 3Digital Solutions and Services, University of Derby, Derby DE22 1GD, UK; 4School of Health Sciences, University of Nottingham, Nottingham NG7 2RD, UK

**Keywords:** cardiac rehabilitation, ethnic minorities, cardiovascular rehabilitation, cardiovascular disease

## Abstract

Cardiac rehabilitation (CR) can improve cardiovascular risk factors, decrease cardiac mortality, and promote healthy lifestyle behaviours. However, services remain underutilized by groups of ethnic minorities. The purpose of the study was to identify patients’ personal CR experiences to identify the differences CR makes towards minorities’ lifestyle. An initial electronic search was performed in 2021 for papers ranging from 2008–2020 across specific databases, including *PubMed*, *EMBASE*, *APA PsycINFO*, *CINAHL* (*Cumulative Index to Nursing and Allied Health Literature*), and *Medline*. *Google Scholar* was also used to supplement the search process and to identify studies performed within grey literature. A total of 1230 records were screened, of which 40 were assessed for eligibility. The final sample consisted of seven qualitative design studies that were identified for inclusion in this review. Based on patient personal experiences, this review identified that ethnic minorities continue to remain disadvantaged when accessing healthcare interventions, primarily as a result of cultural behaviours, linguistic barriers, socioeconomic status, religious and fatalistic beliefs, and low physician referral rates. More research is needed to elucidate this phenomenon and address these factors faced by ethnic minorities.

## 1. Introduction

Cardiovascular diseases (CVDs) account for 17.9 million global deaths each year, with more than four out of five cases owing to coronary heart disease (CHD) [1]. As a result, 80% of premature deaths are due to uncontrolled risk factors such as unhealthy diets, lack of physical activity, and tobacco, which are preventable and treatable [1]. To address this, organisations such as the American Association of Cardiovascular and Pulmonary Rehabilitation (AACVPR) and the National Institute for Health and Care Excellence (NICE) in the United Kingdom (UK) responded to the global epidemic with strategies to implement secondary prevention recommended guidelines encouraging the use of cardiac rehabilitation (CR) services for sufferers of CVD to optimise a healthier lifestyle [2,3]. Plans to tackle the disease has also been echoed in Canada and in Hong Kong, forcing legislators to develop preventative guidelines to help manage the population at high risk of the disease [4,5].

Cardiac rehabilitation is an individualised programme designed to provide secondary prevention therapies after the experience of cardiovascular (CV) injury [6]. It is known to improve quality of life and restore functional capacity while reducing hospital readmissions and preventing the recurrence of cardiovascular events [7]. Annually, in the U.K., 100,000 patients initiate CR programmes [8], reported to cost GBP 477 per person and with an average total consumption of GBP 42 million a year [9]. Studies that have focused on the cost and clinical health benefit of CR identified interventions to be cost effective in heart failure patients [10], naming exercise interventions as producing the most cost-effective strategy in comparison to psychological interventions [11] owing to decreased rates of hospitalization [10]. Rehabilitation programmes comprise of exercise therapy, medication education, psychological support, diet and nutrition, and smoking cessation, all components tailored towards reducing CV risk factors [12]. The metabolic and physiological effects of these programmes offer comprehensive resources that contribute to tackling CVD, which surgical or primary coronary interventions cannot address alone [13]. Interventions are offered to all eligible individuals after the event of a heart attack, acute coronary syndrome, heart surgery, chronic stable angina, percutaneous coronary intervention, valve surgery, peripheral arterial disease, or heart failure [14]. However, despite efforts to promote the use of these services, participation in CR programmes remains underused and underutilised worldwide [15]. Individuals particularly affected are those from ethnic minority backgrounds [16], who are three to five times more likely to be hospitalised with CV risks factor events than the White population [17]. Studies reporting on CV care in the United States (U.S.) identified that the Black, Hispanic, and Asian minority groups are severely under-represented in CR [18] and 7.8% less likely to participate in CR activities than their counterparts (19.6% vs. 20.4%) [19]. Lack of physician referral and financial resources are factors that have previously been suggested as barriers [20] to attending CR.

The literature surrounding ethnic minorities’ experience of CR is limited and poorly understood, which challenges the ability to gain in-depth, resourceful insights to address factors related to poor use of services within these groups. The importance of capturing ethnic minorities’ experiences as communities severely affected by CVD is vast and will help to inform healthcare professionals (HCPs) of ways to improve service-delivery strategies, better understand how to serve ethnic groups, increase participation, raise awareness of CR programmes, and improve health outcomes to effectively address care needs. Therefore, this review sought to explore the experiences of CR faced by ethnic minorities, compare experiences to highlight less-effective CR strategies, and identify factors causing engagement and disengagement with CR activities to address health disparities and illuminate findings within this field to add to the existing literature. In this review, in the interest of economy and parsimony of presentation, we use “ethnic” to refer to the Office of Management and Budget [21] racial and ethnic categories.

## 2. Materials and Methods

An initial electronic search was performed in 2021 for papers ranging from 2008–2020 across specific databases including *PubMed*, *EMBASE*, *APA PsycINFO*, *CINAHL* (*Cumulative Index to Nursing and Allied Health Literature*), and *Medline* [22]. The search was later updated in November 2022, accounting for papers published up to November 2022. This is in accordance with the Preferred Reporting Items for supporting Systematic reviews and Meta-analysis framework (PRISMA) [23].

### 2.1. Search Strategy

The search involved three key electronic databases: *CINAHL* and *CINAHL Complete*, *Medline*, and *APA PsycINFO.* Additionally, *Google Scholar* was searched, with the first 200 studies included in line in accordance with suggestion from Haddaway et al. [24]. Searches ranged from inception to November 2022. Initial search terms were developed with a specialist librarian and were later revised to capture specific interests detailed by ethnic minorities’ experiences. Search terms included a combination of terms related to (1) cardiovascular rehabilitation, (2) ethnic minority groups, and (3) qualitative studies. An example of the search syntax and key terms can be located in the Appendix B. Reference lists of relevant papers were also searched for studies potentially meeting inclusion criteria. Studies were assessed against an adjusted version of the PICO framework [25] in accordance with the inclusion criteria detailed in Table 1. PICO.

### 2.2. Inclusion and Exclusion Criteria

The review searched for qualitative studies that investigated patients from an ethnic minority group who had suffered cardiovascular trauma or disease, with particular focus on participants’ experiences and engagement with CR programmes within a hospital or any community setting. Patients from an ethnic minority group were defined as individuals listed under the Office of Management and Budget racial and ethnic categories [21]. Studies that addressed secondary prevention programs such as behaviour and lifestyle modification, health and psychological intervention, medication education, smoking cessation, diet and nutrition, or physical exercise offered through CR participation were considered in this review as programs documented to deliver manageable healthy behaviours that individuals are more likely to adhere to in order to manage CVD risk factors [26] post cardiac events. Studies documenting home-based CR interventions, did not involve attendees of CR programmes, included participants below 18 years of age, and where the outcome was related to specific genders were excluded from this review. Only studies presented in the English language were included due to not having access to translating software. This process was facilitated by an inclusion/exclusion criteria and screening tool that can be viewed in the Appendix A to guide and narrow down the overwhelming extent of literature [27] when searching for appropriate studies.

### 2.3. Screening and Study Selection

All titles and abstracts of potential studies were independently screened by the lead author (A.C.T.). After removal of duplicates, the search citations were exported into Mendeley reference management software and further assessed for eligibility. Full-text articles were reviewed by two authors (A.C.T. and E.T.), and studies that did not meet the inclusion criteria were carefully excluded with reason. The final included studies were selected through discussion, and any conflicts were resolved by involving another author where applicable (Y.K.).

### 2.4. Quality Appraisal

Studies of suitability included in this review were assessed by the Critical Appraisal Skills Program (CASP) checklist against risk of selection, allocation, attrition, detection, and performance bias [28] and independently scored out of 10 for their quality. This checklist has been well documented and praised for its evaluation of the quality of qualitative evidence published within the healthcare domain [29]. The majority of the included studies were rated as good, with an average score of 7/10. All articles were assessed for their quality with comments. An example of this can be viewed in Appendix A.

### 2.5. Data Extraction

Data were compiled into a table by ACT and independently reviewed by E.T. to identify material relevant to this review. Data included information about the year of publication and author(s), study design, setting and location, number of participants, study aim, type of CR intervention, comparators, participants’ ethnicity, and relevant topics.

## 3. Results

Figure 1 details the study selection process of articles included and excluded during the screening phase at each stage. The database search yielded 19,521 results. Additionally, 17 records were identified through hand searches and references lists. After the removal of duplicates and studies from supplementary searches, 1230 records were screened, of which 40 were assessed for eligibility. The removal of studies with reasons led to seven articles being identified for inclusion in this review.

### 3.1. Studies Characteristics

Table 2 presents the characteristics of the included studies. All studies were of qualitative designs, which utilised face-to-face, semi-structured interviews as data collection methods. Sample sizes ranged from 7 to 65, which included the following ethnicities: The majority of the studies sampled participants were from South Asian/Indian backgrounds (*n* = 108) [30,31,32,33,34]. Astin et al. [30] and Bhattacharyya et al. [31] had comparative studies that collectively involved 30 participants who identified as White European. Koehler et al. [35] reported 7 informants as African American, while Wong et al. [36] identified 22 Chinese participants. Studies were conducted from the U.K. (*n* = 3), Canada (*n* = 2), USA (*n* = 1), and Hong Kong (*n* = 1). A representation of 57 females and 110 males participated in this study. The accumulated sample of 167 participants was included in this review. Study data were independently extracted by the first author and then coded and synthesised in relation to the review’s aims. Codes were grouped into initial themes, which were then discussed and defined in collaboration with the second author. The final findings include three themes: (1) general CR participants’ experiences, (2) factors encouraging CR, and (3) factors discouraging CR.

All studies targeted key aspects of CR interventions involving participants from ethnic minority backgrounds and discussing dietary information, lifestyle, physical activity, medication advice, and recommendations [30,31,32,33,34,35,36]. However, four studies did not sufficiently outline details of the interventions (e.g., the time frame of interventions) [30,31,34,35].

Studies that did report details of the intervention indicated that CR sessions ranged from 3 weeks to 10 months [32,33,36]. Some studies outlined a preference for more short-term interventions. Wong et al. [36] examined the attitudes of Chinese patients toward CR programmes and outlined the importance of using a 6-week program as opposed to a 12-week CR programme to avoid disruptions to patients’ working schedules and to improve adherence. Similarly, Banerjee et al. [32] examined cultural factors facilitating CR participation amongst a South Asian population and reported patients’ preferences for undertaking a shorter, 10-week CR programme. These patients reported that they were more likely to attend shorter health programmes, which would cause less inconvenience to their working schedules.

There were also differences between studies regarding the location at which rehabilitation services were offered. Of the seven included studies that did report on program locations, three reported that sessions took place in hospitals’ outpatient departments [32,34,36], with one study reporting that sessions took place in a leisure centre [31]. Some studies reported interventions as supervised [32,36] or unsupervised [33] classes. Additionally, all but one study [31] discussed CR sessions being led by CR nurses, health professionals, pharmacists, occupational therapists, physiotherapists, and dieticians, which was positively remarked upon by minorities.

### 3.2. Comparison Studies Examining Experiences of White and Ethnic-Minority Patients

Two studies examined the experiences of participants from ethnic minority groups compared with patients who identified as White [30,31]. Both explored the experiences of patients who identified as South Asian in comparison to those who identified as White and reported cultural differences in engaging with rehabilitation.

On attending follow-up appointments, patients who identified as South Asian were more likely to be accompanied by their children, who would assist with language barriers and aid communication with the health care professionals (HCPs) [30]. On the other hand, patients who identified as White were more likely to be accompanied by their spouses [30] and jointly receive first-hand information from HCPs, with the advantage of directly addressing health concerns and acting on new information to improve their health.

In terms of lifestyle changes advised by HCPs or at rehabilitation, there were differences between ethnic groups, such as greater shared food preparation (and therefore greater shared responsibility of modifying diet) in patients who identified as White compared to those who identified as South Asian, who reported more difficulties in changes to their diet, which may cause additional work or resistance in uptake or continuing with dietary modification due to compromises in taste [30]. This was of particular concern reported amongst women who identified as South Asian, who found challenges in modifying their diet to address health needs issues in order to comply with their spouses or families’ dietary preferences. Stress was viewed as an important factor amongst both South Asian and White families. Practical measures to minimise stress experienced by their relatives post cardiac event were more likely to be endorsed by their children than their spouses in South Asian families [30]. A common stress-relieving approach practiced in South Asian families was to withhold or delay information sharing with their relatives relating to their health condition in attempt to avoid inducing stress for their parents.

On the other hand, a very different approach was experienced by patients who identified as White, as they were supported by both their spouses and children to take stress-relieving measures and enjoyed activities such as listening to story tapes together as a form of relaxation therapy to alleviate stress [30].

Similarly, carers and family members of both patients who identified as White and South Asian reported taking supportive roles in encouraging recovery following cardiac events (e.g., myocardial infarction) [30,31]. This included food preparation, exercise adherence, and attending follow-up appointments at hospitals/community centres [30,31].

Both studies indicated shared experiences of engaging in CR regardless of ethnicity. For example, both participants who identified as South Asian and White were motivated to make lifestyle changes. Changes involved reducing the consumption of red meat [31] and accepting foods prepared by spouses with less salt, fat, and sugar content [30]. Although difficulties were reported in Astin et al. study [30] in instances where strategies were implemented by spouses to reduce their husbands’ sugar intake, this was tolerated depending on mood levels.

### 3.3. Factors Encouraging Patients to Attend Rehabilitation Sessions or Engage in Rehabilitation Activities

Across five studies, some participants reported that rehabilitation programmes gave a positive experience, which encouraged them to attend rehabilitation [31,32,34,36]. Studies reported that CR increased patients’ confidence and that they enjoyed meeting peers, as it benefitted their wellbeing [31,34,35]. One study reported the importance of peer interaction [36] contributing to successful engagement and completion of CR programs. Such acknowledgment provided patients with enhanced knowledge of their condition and offered constructive shared dialogue while reducing anxieties [31]. Some patients outlined optimism in achieving health goals, feeling stronger, and obtaining a sense of renewed energy [35]. Such views were reported as attributed to CR nurses’ and HCPs’ emotional and functional abilities to support such groups during rehabilitation classes [32,35].

Family support both in terms of (1) practicality in attending sessions and engaging with patients in CR-related activities and (2) emotional support/encouragement was also a factor encouraging greater reported attendance of rehabilitation. Some patients took great responsibility for attending CR sessions, which involved walking at length to reach their appointments. In such instances, patients reported family members taking supportive roles to prevent lengthy walks in fear of their condition, offering to assist with transporting to and from sessions [32,34], and encouraging attendance. Across several studies, patients reported being more likely to engage in CR-related activities when given instrumental support from partners [30,32,36]. For example, in studies from Astin et al. [30] and Wong et al. [36], participants reported spouses adopting the behavioural changes and activities recommended by CR, for instance, through regular walking together. When family members provided an encouraging and positive attitude about rehabilitation, patients were more likely to be motivated to attend CR [32]. However, some participants outlined that receiving conflicting advice in relation to treatment options or CR-recommended exercise therapy from family or relatives resulted in feelings of confusion and isolation [31,35].

Overall, five studies reported that friend and family involvement contributed to successful completion of CR activities [30,32,33,34,36]. Less-supportive statements appear to have come from African American, Indian, and Bangladeshi participants, who expressed a lack of support from family members, which was conflicting or given grudgingly [31,35].

### 3.4. Factors Discouraging Patients to Attend Rehabilitation Sessions or Engage in Rehabilitation Activities

Across all studies, participants faced socioeconomic, religious, linguistic, and in some cases healthcare and physician referral issues as barriers to engaging in CR programmes [30,31,32,33,34,35,36]. These factors caused patients to take matters into their own hands regardless of whether they were led by their religious faith or supported by family members or friends to seek out health opportunities. For example, in the Koehler et al. [35] study, patients who had learnt about the CR program through friends or family but had not received an appointment to attend would approach their physician directly to initiate a referral to enrol in classes.

Financial concern was reported, particularly among patients from low-income backgrounds, Indian/Bangladeshi and African American [31,34,35] backgrounds, and was linked to inability to attend CR due to transportation costs and no parking [34]. These patients also reported being unable to afford medication, therefore leading to poorer adherence to medication [35]. Financial concerns were also related to low mood changes regarding daily routines and a general lack of activity as symptoms of depression evolved, making adherence to CR-related activities more difficult [31].

Across four studies, religion was also reported as a factor that could discourage patients from engaging fully in CR activities amongst South Asian/Indian Bangladeshi and African American patients [31,33,34,35]. Interestingly, while faith and fatalistic views supported patients’ recovery from cardiac events, it was also highly regarded over medical advice received from HCPs [33], which led patients to make important lifestyle decisions about their health and recovery based on their own religious views [34]. Dilla et al. [33] reported that dietary, exercise, and medication timing that had a profound effect on patients’ religious rituals was avoided, as such activities disrupted attendance of their place of worship or did not conform with their faith beliefs. Some patients reported that their health injury had been caused as a results of bad dietary and lifestyles choices, and it was by fate that such events had occurred [31,34]. This expressed belief led patients to believe that regardless of the health information they received from HCPs, the direction of their health was in their deities’ hands, which could not be controlled [37].

One of the most important barriers to attendance and engagement in CR activities was poor communication between HCPs and patients. Of the seven reviewed studies, six studies [30,31,32,33,34,35,36] reported on the challenges patients experienced to access rehabilitation services not offered in their mother tongue. This prevented meaningful dialogue between HCPs and patients, limited understanding of health conditions, and involved other forms of communication such as interpreters to address linguistic barriers [30,34]. Two studies referenced the use of interpreters [30,31]. One reported family members being utilised to interpret information [30], while the other study documented interpreters employed by the National Health Services in the U.K. [31]. Although efforts were made to address language barriers, patients reported employed interpreters to be ineffective when assisting in clinical appointments.

Lastly, the referral system to rehabilitation services via physicians and HCPs was considered a barrier to attendance of CR and engagement in CR-related activities. This was identified in two studies [31,35] as commonly expressed by African American and Indian and Bangladeshi patients. Both studies documented patients’ non-referrals to CR while also reporting patients being unaware of the services that rehabilitation provided. Eight patients admitted being uninformed of such services [31], while two patients admitted receiving no referral at all [35]. This indeed provoked frustrations and disappointment among patients who had voiced concern for the services to be promoted and encouraged post discharge from the hospital. These views were not expressed amongst other participants who identified as White or Chinese patients, as they were more likely to receive a referral to CR programmes.

## 4. Discussion

This review synthesised qualitative evidence identified in grey and published literature pertaining to ethnic minorities’ experiences of CR interventions. Further, attempts to gather literature suitable to the research question was also performed by seeking expert consultation. However, the quality of data recommended was insufficient or lacked relevance to the review; hence, the number of articles included in this scoping review is small. To avoid any unclarity, the value of the review’s main findings are emphasised in this discussion to unveil current, lived experiences by ethnic minority groups and to seek ways to improve those experiences. Additionally, our results highlight that ethnic minorities appear to remain disadvantaged when accessing interventions, primarily as a result of their cultural background, linguistic barriers, and low physician referral rates, amongst several other concerning factors mentioned earlier in this review. Consequently, these findings have been reported within the current literature among other authors who have expressed similar concerns [38,39,40,41] and have given recommendations for system- and patient-level developments to be considered for integration within healthcare fields. However, little research has been published on the success of these recommendations for minorities.

The structures of the CR programmes offered to minorities within this study were found to be a deciding factor for whether patients attended and how convenient programmes were for them. The outpatient views expressed from minorities were dependent upon (a) duration of CR programmes, (b) location of where rehabilitation was performed, and (c) whether interventions were supervised or unsupervised. Most patients favoured short-term programmes, as they were more likely to attend CR sessions that caused less of a disruption towards working and lifestyle environments. It may be suggested that in order to increase minorities’ compliance of attendance to CR interventions, specialists should consider promoting shorter programmes within safe remits tailored towards patients’ preferences [42,43] while increasing awareness of the benefits of CR. Moreover, the health benefits of a shorter programme are far more significant than non-attendance, which leads to a better quality of life, exercise tolerance, and good mental health status [44]. Additionally, programmes that were clinically led by HCPs were more likely to be appreciated by ethnic minorities, as the presence of professionals created a safe space for patients to be able to exercise within supervised environments [32,37,45]. These factors should be highlighted for future rehabilitation services to offer such groups clinically led sessions as a first point of call to increase success rates of compliance and achieve positive health experiences. Further, as a response to alternative measures designed to increase CR participation, such as home-based CR, a programme to provide remote training with indirect education and exercise supervision [46], may not be suitable for these ethnic groups, as they are least likely to take part in indirect CR sessions.

One major factor revealed in this review by participants who identified as South Asian was communication and language barriers. Given that most patients in the presented studies were non-English native speakers and were in programmes led in English-speaking countries, this immediately puts patients at a greater risk of disadvantage than their counterparts when accessing services insufficiently built to accommodate communication and language barriers [47,48]. Endorsing family members to act as translators during appointment visits may not be the most suitable method, as this can create further confusion and cause tension between patients and their relatives [30]. Additionally, the lack of knowledge of medical jargon and understanding from family members makes them the least-appropriate candidates to translate health information [49]. As other authors have suggested, services must continue to consider adapting multicultural pathways and investment in expert translators within health fields [50] to eliminate ineffective communication experienced by minorities. Further, well-designed written information presented in patients’ own language may help to reduce communication barriers and to improve engagement, patient’s health knowledge, and HCPs interaction [51]. These outcomes should be continuously considered within CR programmes and reinforced to promote the best patient healthcare experiences.

Challenges in adopting healthier dietary choices were more difficult for participants who identified as South Asian than White. These findings are in keeping with the current literature, which supports views that dietary adherence amongst South Asian patients remains poor and inadequate, with diets containing high consumptions of trans fats, simple sugars, sodium, and foods high in cholesterol [52], which contribute to cardiovascular disease. To address these issues, studies investigating dietary habits among South Asians in healthcare have strongly recommended [52,53] culturally appropriate interventions aimed at promoting healthier dietary food selections to reduce CV health factors and improve healthier outcome choices for this community.

Both participants who identified as South Asian or White were motivated to make lifestyle changes to improve health risks and appreciated supportive roles from their spouses. However, participants who identified as South Asian required more encouragement in accepting changes to their lifestyle and diet, proving more of a challenge for South Asian spouses to implement new ideas to support their partners’ health needs. This outlines the notion that the support and advice promoted in CR programmes should also be reinforced and extended to family members and spouses who face challenges to support their relatives in making changes to enhance lifestyle experiences [54,55].

This review also identified the difference that peer, family, and HCPs support made to minorities’ experiences when engaging in CR activities as voiced by participants who identified as South Asian, Chinese, or African American. Peer and HCP support played a significant role in encouraging engagement and successful completion of CR activities. The ease of conversation amongst patients who shared similar health experiences provided motivation and reassurance to manage health conditions, which was regarded as an important enjoyable factor. Additionally, HCPs who were able to identify with groups of minorities to provide constructive cultural supervision to optimise patients’ health were praised. Such facilitators are known advantages [56] considered to promote a more favourable experience with rehabilitation programmes. Moreover, practical and emotional family support also contributed to successful engagement with CR programmes. However, reinforcement is necessary by specialists on commencement of programmes, as some families (African American, Indian, and Bangladeshi) may require additional coaching compared to others [57]. Furthermore, this presents an opportunity for specialists to work in partnership with family members to communicate ideas to achieve better health experiences for their loved ones post cardiac injury.

Consequently, the data captured factors linked to disengagement in rehabilitation activities experienced by minorities related to socioeconomic, religious beliefs, linguistic, and physician referral errors. Financial concerns were expressed amongst Indian/Bangladeshi and African American patients as limiting participation in CR sessions. These factors evidently increase non-attendance to CR, which could be remedied by offering incentives that potentially decrease cardiac health risk factors and increase CR participation when offered. Gaalema et al.’s clinical trial findings [58] investigating patients from lower-socioeconomic status (SES) populations (Black African American, Indian, and Hispanic) unveiled evidence to suggest that financial incentives improve completion and participation in CR programmes. Further, Mathews et al. [59] suggested that rehabilitation programs should offer assistance with co-pays and out-of-pocket expenditure; for example, medication costs and transportation vouchers are likely to improve adherence to CR once these barriers are addressed. There is a need for CR specialists to take into consideration patients’ SES during the referral phase and work towards measures to prevent barriers associated with patients’ personal circumstances that limit access to care facilities and treatments.

Religious and fatalistic beliefs were considered priorities and in some cases, regarded above health interests amongst patients from ethnic minority groups. This may indicate the need for culturally appropriate and sensitive information to be incorporated within CR programmes in hopes of supporting patients’ health process and improving attendance. Moreover, this will give CR specialists the opportunity to understand how religious and fatalistic beliefs aid recovery post care and use innovative ideas to help benefit other patients who also share similar beliefs. Further, the literature emphasises the need for religious strategies to be integrated into CR programs to assist health outcomes [60], which may require developing individualised care plans aimed at improving patients’ experiences and adherence to rehabilitation.

Physician referral is considered the gateway for participation in CR activities for all eligible patients post cardiac trauma [61]. Most referrals are initiated prior to patient discharge from hospital; however, this review revealed inconsistencies across some groups of ethnic minorities. Absent referrals were identified amongst African American and Indian and Bangladeshi patients. Previous studies have also expressed similar concerns of patients not receiving CR referrals [62], which may indicate the lack of in-patient health education received by these groups. Such findings are worrying and could evidently lead to further disparities faced by ethnic minorities. Patients left unaware of treatment options run the risk of adopting unhealthy lifestyle habits and increasing cardiac risk factors. Studies have suggested ways to overcome barriers to patients’ experiences, suggesting strategies to increase enrolment rates by implementation of automatic referral and patient liaison officers [63,64] to provide bedside education to all eligible patients regarding the benefits CR has to offer post discharge from hospital.

This review has highlighted the experiences of a number of ethnic minority groups. However, population diversity transcends ethnicity and varies immensely globally. These findings are limited to the ethnic group categories selected by the researchers through their data collection tools. It is important to recognise that there were no studies that included participants who identified as having a multiple ethnic heritages or were able to give a self-definition. This excludes a large section of ethnic minority groups with valid service experiences, which ineffectively limited a more robust comparison between ethnicities. Additionally, the way in which ethnic groups are categorised can hide the huge heterogeneity within groups. For example, broad terms in the USA and the U.K. denoting African descendants, such as African American and Black British, ignore religious differences. Moreover, the ethnic group category of White may exclude national minority groups such as Gypsy or Irish Traveller, meaning that poorly designed data collection tools used in research can weaken the value of ethnic categorisation for understanding disparities in health. Further limitations were also recognised by this review, as the literature was limited to the English language. Secondly, a more in-depth analysis would be helpful; however, the feasibility of such an investigation was uncertain before the study was performed. Based on the findings from this scoping review, a more robust form of study such as systematic review is needed about this topic. Thirdly, because all authors are researchers in health and education, author bias might have been present.

## 5. Conclusions

Insights gathered from this review regarding patients’ personal CR experiences highlight that minorities appear to remain disadvantaged when accessing healthcare interventions, primarily as a result of their cultural behaviours, SES, linguistic barriers, religious and fatalistic beliefs, and low physician referral rates. Strategies to improve patients’ CR experience are warranted, which may require a more individualised approach to address health concerns.

However, the effectiveness of how some interventions could be incorporated to CR programmes remains to be evaluated due to a lack of rigorous evidence and clinical trials involving ethnic minorities. More research is needed to examine this phenomenon and address these factors faced by ethnic minorities.

## Figures and Tables

**Figure 1 healthcare-11-00757-f001:**
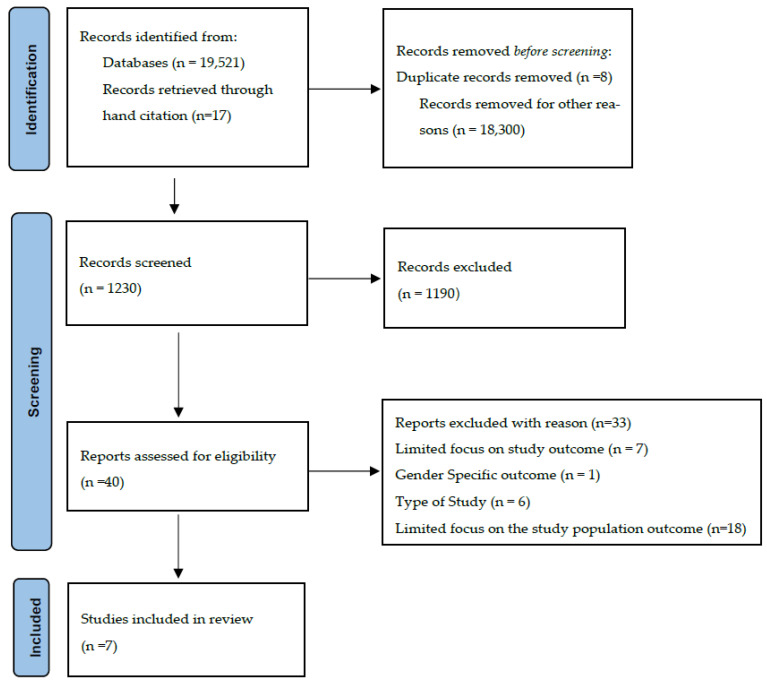
Study Selection Process.

**Table 1 healthcare-11-00757-t001:** PICO.

Population	Male and female ethnic minority patients: individuals of American Indian, Asian, Black or African American, Hispanic, and White descent having origins in any of the original peoples of Europe, the Middle East, or North Africa, as defined by the Office of Management and Budget to the Standards of the Classification of Federal Data on Race and Ethnicity (1997) aged 18 > who attended any cardiac rehabilitation interventions at any time and experienced any clinical cardiovascular trauma/disease in their lifetime.
Intervention	All cardiac rehabilitation programs, i.e., behaviour and lifestyle modification, health education and psychological, therapy, or physical exercise given for the care and recovery of any cardiac event that requires cardiac rehabilitation.
Comparison	NA
Outcome	Qualitative views or perceptions of attending cardiac rehabilitation; positive or negative personal or group experiences of attending CR services.

**Table 2 healthcare-11-00757-t002:** Studies Characteristics.

Author and Year	Country	Study Design	Sample Size	Sample Participants Ethnicity	Male: Female Participants	Relevant Themes	Quality Assessment
Astin, et al. (2008) [30]	UK	Qualitative	65	South Asian & White-European	36:29	Provision of advice and information, exercise, and Dietary change,	7/10
Banerjee 2010 et al. (2010) [32]	Canada	Qualitative	16	South Asian	13:3	Predisposing factors -Patients felt safe when performing exercise in sessions Pts had positive views of CR from relatives Positive education of physical activity & nutritional info Enabling Factors- Accessible Transport, Flexibility of CR Sessions Physician Referral Reinforcing Factors- Family support, physician support and FU, caring and support staff, experiencing positive outcomes.	7/10
Bhattacharyya, et al. (2016) [31]	UK	Qualitative	28	White British, Indians and Bangladeshis	23:5	Low mood, anxiety and fear, perception of self and how others perceive them, Physical Impact, attitudes towards the future, Cardiac rehabilitation, Social and professional support, lifestyle changes and return to work	7/10
Dilla et al. (2020) [33]	UK	Qualitative	14	South Asian	9:5	Dietary Information Exercise Religious Faith	8/10
Galdas et al. (2010) [34]	Canada	Qualitative	15	South Asian	10:5	Dietary advice Religious Faith ongoing interaction with peers- offered psychological support	8/10
Koehler et al. (2020) [35]	United States	Qualitative	7	African American	4:3	Religious Faith Physician Referral Participants experienced inconsistent referral to and utilization of Cardiac Rehabilitation Barriers to CR was discussed	7/10
Wong et al. (2016) [36]	Hong Kong	Qualitative	22	Chinese	15:7	Participants attitude toward OCRP, which included informant perception, affection, and practice Exercise Behaviour – intention and maintenance	7/10

## Data Availability

Not applicable.

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
