# Peer review of "Ethnic Minorities’ Experiences of Cardiac Rehabilitation: A Scoping Review"

_healthcare, 2023, doi:10.3390/healthcare11050757_

Round 1

Reviewer 1 Report

This review collected 7 qualitative design studies, describing reasons why  ethnic minorities are not receiving the same extent  cardiac rehabilitation than the native whites, withdrawing the conclusion from only 167 patients in 7 studies, which seems to be short of data support. The conclusions are obvious and can be predicted. 

I don't think the article is valuable enough to be published.

Reviewer 2 Report

Ethnic minorities experiences of cardiac rehabilitation: A scoping review

The main objective of this scoping review was to explore the experiences of cardiac rehabilitation faced by ethnic minorities. The other aims were to compare experiences and identify factors causing engagement and disengagement to rehabilitation activities in order to address health disparities.

I have few comments about the manuscript.

1.     Could the authors describe the cardiac rehabilitations and literature behind them in more detail? The current text reads: ”Studies which addressed CR interventions as CR programs delivering behaviour and lifestyle modification, health and psychological medication education, therapy or physical exercise, given for the care and recovery of any cardiac event which required CR were considered.” Are all these rehabilitations effective in secondary prevention of cardiac events? What about cost-effective? How do they differ from each other in regards of effectiveness? 

2.     I missed more rationale behind the research question. Why is it important to study the experiences of ethnic minorities? Is there literature showing that bad experiences are related to less effective results of the rehabilitation? How does this relate to the health outcomes of ethnic minorities after cardiac rehabilitation?

3.     There is a tiny confusion in the use of term ”cardiovascular disease”. In the beginning of the introduction, it refers both to cardiac diseases and strokes. However, it seems that only rehabilitation of cardiac diseases was considered. Could you please clarify if you included studies on stroke or other cerebrovascular disease rehabilitation and why (not)?

4.     The study describes well the experiences of ethnic minorities. Would it be possible to add more comparison between ethnicities? How do the experiences differ from each other between them?

5.     There was some repetition in the manuscript as the main findings were mostly repeated in the discussion section. Would it be possible to shorten the discussion and keep the focus on methodological discussion, future research prospects, clinical implications etc, and to avoid repetition?

Reviewer 3 Report

The predominance of national minority groups is increasing every year. It is gratifying that the authors have devoted their review to this problem. The low level of cardiorehabilitation in this category of patients is influenced by numerous factors, such as communication barriers, diet choice, the cost of drugs, lack of awareness of medical services, which leads to a deterioration in the quality of life, a decrease in exercise tolerance, a poor mental state, etc.

The review presents extended data and indicates possible ways to solve the problem.

The review is written in good English, and the bibliography contains 50% of articles no older than 5 years.

There are some minor comments on the text.

Not deciphered:

US – line 50;

National Health Services – line 271 and NHS – line 272;

SES - line 388, later on Socioeconomic status.

Health Care Professionals transcribed by (HCP) (line 178), then as HCPs, again HCP on the lines 318, 332, 362.

There is no year of publication of Article 54 (line 589).
